# The Temporary Mental Nerve Paresthesia as an Outcome of Dentigerous Cyst Removal during Preparation for Dental Implant Placement: A Case Report

**DOI:** 10.3390/medicina59040711

**Published:** 2023-04-05

**Authors:** Kamil Nelke, Maciej Janeczek, Edyta Pasicka, Krzysztof Żak, Marceli Łukaszewski, Radosław Jadach, Maciej Dobrzyński

**Affiliations:** 1Privat Practice of Maxillo-Facial Surgery and Maxillo-Facial Surgery Ward, EMC Hospital, Pilczycka 144, 54-144 Wrocław, Poland; 2Academy of Applied Sciences, Health Department, Academy of Silesius in Wałbrzych, Zamkowa 4, 58-300 Wałbrzych, Poland; 3Department of Biostructure and Animal Physiology, Wrocław University of Environmental and Life Sciences, Kożuchowska 1, 51-631 Wrocław, Poland; maciej.janeczek@upwr.edu.pl; 4Department of Anaesthesiology and Intensive Care, Sokołowski Hospital, Sokołowskiego 4, 58-309 Wałbrzych, Poland; 5Dental Salon Privat Dental Office, Horbaczewskiego 53a, 54-130 Wrocław, Poland; 6Department of Pediatric Dentistry and Preclinical Dentistry, Wrocław Medical University, Krakowska 26, 50-425 Wrocław, Poland

**Keywords:** mandibular cyst, xenograft, mental nerve, case report, bone reconstruction

## Abstract

The usage of xenograft material is widely used in almost all oral cavity bone defects for regenerative and reconstructive purposes. The presented xenograft usage in the following care report enabled good bone defect healing and enabled the preservation of affected premolars. It is quite common to use any possible variations of bone materials to ensure bone defect improved healing. In some cases, the scope of surgeries requires the removal of each cyst within close proximity to various nerves and vessels. The inferior alveolar, infraorbital, lingual, and mental nerves are those most commonly adjacent to most operating sites in jaw bones. The usage of some additional materials such as collagen sponges, bone substitutes, resorbable membranes, or other additional materials are useful in each bone defect reconstruction but should be handled with care, as described in the following case. Before planning their usage, it is important to perform each surgery with close cone beam computed tomography imaging, which is very helpful to establish the scope of each lesion and the proximity of vital structures. There are a lot of factors that might influence any possible nerve damage, especially the different nerve anatomical variations. Even factors including the subperiosteal preparation and compression of adjacent tissues might influence later nerve function. When the lesion is expanding through the buccal cortical plate and when soft tissue fluctuation is present, some special care is needed. Similar to the presented case, a limitation in crushing, blowing, or any irritation of nerve fibers improves later postoperative outcomes. When the wound and surrounding tissues are handled with care, a limited possibility of any damage or paresthesia can occur. When the nerve itself is damaged or cut, loss of function can be permanent. Immediately after or even prophylactic prescription 1–2 days before the surgery of Vitamin B with NSAIDs (Non-steroidal anti-inflammatory drugs) (or other additional supplementary medicaments can improve nerve function in time. Possible nerve damage can be divided into many etiological factors. A quite different situation arises when the nerve is pulled in by the cyst growth into the cyst wall. The presented case report describes the outcomes of a cyst removal from the mandibular basis and treatment modalities.

## 1. Introduction

In the oral cavity, the main innervation of jaw bones is related to the second and third branches of the trigeminal nerve (CNV). The mental nerve (MN) is a sensory nerve from the posterior trunk of the mandibular nerve (CNV3). It provides sensation to the front of the chin and the lower lip, the gums of the anterior mandibular teeth, the area of the chin and skin, and the mucous membrane of the inner cheek. Problems with the nerve cause the chin, the area of the mouth angle, and part of the lower lip on the affected side numbness (loss of sensation) [1,2]. This situation is related to the scope of mandibular body anatomy, where, just under the lingula of the mandible, the mandibular foramen is located with the inferior alveolar nerve passing to the canal, and the lingual nerve, which descends more medially to the floor of the mouth contained in the structures above the mylohyoid line until the gonial tubercules at the mandibular symphysis area [3]. On the other hand, in the middle of the mandibular body, between the premolar mandibular teeth, a mental nerve foramen is present. In normal conditions, these nerves arise from the mental foramen located inferiorly between the roots of the mandibular premolars. The location of the foramen has some anatomical variations as well. After leaving the mental foramen, the MN divides into three main branches beneath the depressor anguli oris muscle, where one branch descends directly to the chin skin area and two others ascend in the skin and mucous membrane of the lower lip. Some additional nerve fibers can also be seen. These branches communicate and anastomose freely with the facial nerve (CNVII, marginal mandibular and cervical branches) and some fibers of the buccal nerve (CNV3). Accessory mental foramen (AMF) can also be present, along with various nerve fibers (Figure 1) [1,2,3,4,5,6].

Possible damage of the mental nerve can be caused by various factors, such as trauma (mandibular fracture); endodontic treatment of mandibular premolars and surgery close to this premolar area (iatrogenic factors (cyst removal)); periapical cyst, abscess, and inflammatory/periapical infections; tumor spread (oral cancer, leukemia, lymphoma (Vincent Sign, tumor spread into the nerve structure)); orthognathic surgery (BSSO-Bilateral Sagittal Split Osteotomy/genioplasty, chin wing approaches); damage to the nerve after abscess drainage; lower lip mucocele removal/labial minor salivary gland retention cyst (nerve fiber damage); local anesthesia administration; compression of the nerve and adjacent tissue pulling; or others [4,5,6,7,8,9,10]. These factors might cause transient or permanent paresthesia. Mental nerve paresthesia (MNP) is reported to be a cause of dental treatment (dental implant, trauma, endodontic approaches) in approximately 15–20% of cases. On the other hand, according to the Chai et al. study, because of growing interest in the transoral endoscopic thyroidectomy vestibular approach (TOETVA), many studies indicate that because of the following, in order to minimize MN damage, special anatomical landmarks should be used [8,9,10,11].

The type and scope of nerve damage can be described by many classifications. Some injuries to any nerve can be easily described by Seddon’s (1942) nerve damage classification, Sunderland’s (1951) or Lundborg’s (1988), and Mackinnon’s (1989). The etiological factors, such as mechanical, chemical, biological, iatrogenic, or other factors, influence the nerve condition and its later degree of regeneration. As reported by Chhabra et al., the Seddon classification is mostly used by electrophysiologists, while the Sunderland grading is more often used by surgeons. Seddon nerve injury is divided into neuropraxia, axonotmesis, and neurotmesis, while the Sunderland classification describes Grades I-V of nerve injuries. Both Grade I and neuropraxia are considered non-degenerative injuries, while the rest are degenerative ones with poorer outcomes [11,12,13,14]. The MN scope, and etiology of its damage, influence the type of nerve damage, which can be either transient hypoesthesia, neuropathic pain, trigeminal neuralgia, or partial sensory loss in different anatomical sites. When present, this can be called numb chin syndrome [11,12,13,14,15,16].

CBCT (cone beam computed tomography) is nowadays a golden standard for the evaluation of any changes in the mandibular and maxillary bones [17]. In cases of cysts and tumors in the bone, not only the shape, size, and diameter of the lesions should be evaluated but also the close proximity of nerves, such as the infraorbital nerve (ION), the lingual nerve (LN), the mental nerve (MN), and the inferior alveolar nerve (IAN) should be closely evaluated. The pathological spread of any lesion in the nerve area might cause some potential nerve damage during any surgery in the jaw bones. Before scheduling any surgical intervention, a good diagnostic approach is mandatory, along with full patient awareness of possible transient or permanent nerve damage. Authors would want to point out their early pharmacological protocol, which can be improved and scheduled even 1–2 days before any major surgery with a close proximity to any nerve structure in the jaw bones.

The following case describes a patient with transient mental nerve injury, who had an instant presurgical pharmacological protocol implemented to affect the possible mental nerve deficiency after the removal of a big radicular cyst in the left mandibular basis as preparation for dental implant procedures.

## 2. Case Description

A 56-year-old patient was referred for consultation because of a cyst in the left mandibular body that was discovered accidentally during CBCT screening before a dental implant placement. An additional clinical survey chart was scheduled to exclude any other diseases that might influence future outcomes, healing, or complications. Because of the bone defect size, a GBR (guided bone regeneration procedure) was scheduled. The presented case report was written according to CARE (case report guidelines) [17,18,19,20,21,22].

The cyst had well-defined borders and was situated between two left mandibular premolar roots. The entire left mental nerve (MN) was surrounded by the cyst, while the inferior alveolar nerve was situated just inferiorly from the cyst (Figure 2). The alveolar ridge remained intact. Two teeth, the 34/35, were stable in the bone without any pathological movement. Because of their lack of vital stimulus to cold, an endodontic treatment was planned. Further extraoral examination revealed no changes within the lymph nodes nor other adjacent tissues. The function of the trigeminal nerve was normal. Firstly, some hygienization and scaling were scheduled.

An intraoral examination revealed some swelling near the 34/35 teeth, with slight fluctuation but without crepitation, and a presence of a purulent fistula close to the vestibular surface under the old prosthetic bridge in the area of missing 36 teeth (Figure 3 and Figure 4). The close radiological and clinical evaluation indicated that it was really a granulation tissue with some inflammation. Because this was the first source of an active infection, it was scheduled for an excisional biopsy with curettage (Figure 5).

Clinical anamnesis and a clear significant physical examination (PE) revealed additional hypertension and hypothyroidism, which had been under pharmacological control. No other genetic, hereditary, psychosomatic, or other disease or disturbance was noted. Familiar history and past surgical intervention history were unrelated to the current condition. The patient was in an overall good clinical status.

After the preparation of the 34/35 teeth and good endodontic treatment, the granulation tissue was first removed. The histopathological evaluation confirmed the presence of granulation tissues with inflammation. During the same visit, two loosened dental bridges on both sides of the maxilla were removed, along with the 13, 23, and 26/16 teeth because of their poor periodontal status, inflammation periapical lesions, and their pathological mobility in the maxillary arch. After a month, when no signs of infection were noted, the cyst removal was scheduled. Further dental implant treatment was planned after the cyst and other sources of dental inflammations would be removed and tissues would be properly healed (Figure 6 and Figure 7).

The main surgical approach was scheduled under local anesthesia and consisted of 3ampules (1.7 mL each) of Ubistesin Forte (articaine with epinephrine, 3M, Maplewood, MN, United States). Approximately 60 min before the procedure, the patient took intraoral premedication consisting of 2 g Amotaks (Polfa Tarchomin, Warsaw, Poland), 400 mg Ibuprofen (Polfarmex, Kutno, Poland), and 1 g of Paracetamol (Biofarm, Poznań, Poland). Both the skin and oral cavity were rinsed with Octanisept solution (Schulke, Hamburg, Germany). Surgical draping U-shaped was used (Matopat, TZMO, Toruń, Poland). To ensure good mouth angle and lips lubrication, a sterile Vaseline (Unilever, London, UK) cream was used. The initial incision was made with blade No 15c B (Swann Morton, WR Swann, Owlerton Grn, Hillsborough, Sheffield, UK). A typical horizontal incision approximately 2 mm below the mucogingival fold was used. The horizontal incision was made in the distance between teeth 33–36. The superior part of the flap was not elevated to maintain any contact with the previously treated tissues near 36 area and gingival sulcus surrounding the 34/35 teeth. A Minnesota retractor (Hossa International, Medinstruments, Warsaw, Poland) was used to ensure good visibility. A full-thickness mucoperiosteal flap was elevated inferiorly and in the medial aspect to ensure the subperiosteal elevation until the mental foramen was identified. The Obwegeser Periosteal elevators (Obwegeser 38-630-06-07-38-630-11-07 17.5 cm/6 7/8”, KLS Martin, Tuttlingen, Germany) ensured the mental nerve safe retraction with the adjacent mucoperiosteal flap. With the usage of a surgical burr (Lindemann, Aesculap, Tuttlingen, Germany), the thinned buccal cortical plate was removed and the cyst cavity was identified. Surgical curettage and bone debridement were performed. Both tooth apexes of the left mandibular premolar were cut with a surgical burr. Inferiorly, a part of the mandibular canal was visible. Slight bleeding from the IAN canal was secured with BloodSTOP hemostatic dressing (Life Science Plus, Mountain View, San Jose, CA, USA). A total of 1 g xenograft bone (1 g The Graft 0.25–1 mm, Manufacturer Purgo Biologics Inc., Korea) and resorbable collagen membrane (15 × 20 mm-OsseoGuard Membrane- Zimmer Biomet, Collagen Matrix Inc., Oakland, CA, USA) was used, while the bone substitute was mixed with two iPRF tubes. Centrifuge (Model: MiniFuge, AMD Company, New Delhi, India) was used at 2660 rpm for 12 min. Careful packing on the bone to maintain adequate pressure on the hemostatic dressing helped maintain the bleeding. The MN was repositioned inferiorly with care when the collagen membrane was placed subperiosteally from the superior and inferior aspects of the bone defect. A control CBCT evaluation confirmed a good final surgical outcome (Figure 8). The wound was sutured with 4-0 interrupted vertical mattress sutures (Dafilon, B Braun, Aesculap AG Am Aescu-lap-Platz, Tuttlingen Germany). The healing period was uneventful; however, the MN temporary MN paresthesia was present for about 6–7 weeks (Figure 9).

The patient continued intraoral antibiotics consisting of Amoxiclav Quick Tab (2 × 1 g—Sandoz Poland, Warsaw, Poland) for seven days and NSAIDs: Ketonal (2 × 0.1 g—Sandoz Poland, Warsaw, Poland) for 3 days, followed by Dexak Sl 0.025 mg three times a day for 5 days (dexketoprofenum, Berlin Chemie Menarini, Berlin, Germany) with Controloc 0.040 mg once a day (pantoprazolum, Takedda Pharma, Tokyo, Japan) for 10 days and local wound care with: 0.1% CHX (chlorhexidine gluconate) (Eludril, Pierre-Fabre Oral care, France) and Alfa Implant Care Mouthwash solution (Atos MM, Warsaw, Poland) was carried away until the 4th week post-operatively. Additional Neurovit 0.32 mg was scheduled two times per day for fifty days in total (1 tabl. Consists of 100 mg wit. B1, 200 mg wit. B6, 0,2 mg wit. B12; 100 tabl.; G.L. Pharma, Gerot-Lannah, Wien, Austria) in the postsurgical period for nerve healing stimulation immediately after the surgery, approximately 4–5 h after the procedure. If the cyst shape, position, and nerve impact is significantly bigger and radiologically relevant, the authors advise to start administration of Vitamin B and NSAIDs 2 days prior to surgery if nerve damage is highly possible. Final outcomes after 1.5 years from surgery are satisfactory with good full oral cavity dental implant rehabilitation procedures (Figure 10 and Figure 11). Further patient prognosis is overall very good and other follow-up appointments have been scheduled.

The MN function in the post-operative period started to improve at the 6th week. Some tingling and stinging was present in the skin of the chin and left mouth angle, while the lip remained numb. The first signs of sensitivity spread in the next 12 weeks and resulted in a total nerve regeneration. Full function was visible at the middle of the 3rd month postoperatively, both in the skin, oral mucosa, lip, mouth angle, and chin area.

## 3. Discussion

Every cyst removal from the jaw bones requires careful planning and CBCT evaluation. Quite commonly, when additional bone grafting or dental implant procedures will be scheduled simultaneously, the patient should be free of any possible inflammation tissues and one of the various possible dental hygienization protocols should be implemented. Furthermore, when the oral cavity before each surgery is also rinsed with an antiseptic solution, it also improves later outcomes. Any potential inflammation present in the oral cavity influences future dental implant healing [14,15,16,17,18]. Each clinician should carefully plan a step-by-step treatment protocol. This can be achieved by various methods and mostly is related to properly scheduled dental treatment consisting of conservative, periodontal, surgical, and other related procedures to improve the dental status of each patient. It is already known and widely established that any bone surgery in the jaw bones should be scheduled when no active source of inflammation is present in the oral cavity [15,16,17,18,19,20,21,22,23]. This situation is also quite important, because, if teeth are not free of plaque and bacteria, or even the same can be associated with inappropriately endodontic treated teeth, whose poor dental status might promote inflammation in the operating field and might cause the necessity of secondary surgery close to the operated site and cause even more possible serious damage to the nerve located in such close proximity.

A close anatomical nerve exposure during the procedure reduces its possible damage or palsy. Intraoperative nerve monitoring could be advisable to reduce this damage and to track the MN, especially during TOETVA surgery, while a normal cyst removal from the oral cavity does not require such a method [20,21,22]. When each nerve is visible and widely exposed, its neuroprotection is more efficient. Factors such as compression, stretching, elevation, and nerve flexion during subperiosteal elevation might lead to MN function deficiencies. When the MN is visible, it is more exposed to any handpiece drills, surgical burrs, and saws; this requires one of the techniques used for its protection. Chhikara et al. described a nerve shield protection technique for different mandibular surgeries [21,22,23]. Similar methods for protection can be used when a periosteal elevator or a Farabeuf or Minnesota retractor is bone based to secure its stable position and avoid unnecessary lateral movements and inappropriate tissue compression in the MN area [20,21,22,23,24,25].

Each surgery in the mandibular located between the premolar region should be made with care. Dentigerous cyst enucleation is the most common procedure. A cystectomy procedure sometimes can be combined with bone curettage or can be used as a second approach after a big cyst marsupialization procedure. The scope of many surgical procedures is still widely discussed. Each surgical approach can be related to various factors such as surgeon preferences, the age of the patient, the size of the bone defect that will heal on its own, will require GBR procedures, or can be treated with marsupialization, overall patient good cooperation, the biology of the cyst/tumor, and its histopathological result from a biopsy or other factors [25,26,27].

Regardless of the type of surgery, each can be burdened by some percentage of possible nerve damage and MN palsy. The discussion on possible oral supplementation for nerve rehabilitation is a topic of many studies. In nerve rehabilitation, a lot of attention is focused on vitamin B12, acetyl-L-carnitine (ALC), adenosine triphosphate/ATP, and others [27,28,29,30]. According to Hasegawa et al.’s study on refractory hypoesthesia, the authors highlight that early oral administration of ATP/vitamin B12 should be used in all cases of hypoesthesia. Presented neurosensory deficits tend to improve in time, within 6 months or less. On the other hand, some authors advise some nerve stimulation, physiotherapy, or other agents [27,28,29]. Moreover, the type of nerve damage influences its healing and future outcome. The Mensink study highlights that chisels and sharp burrs tend to damage more nerve tissues and decrease their healing [30,31,32,33]. The novelty of this case presents that early post-operative or even early pre-surgical intervention pharmacological treatment might improve overall nerve healing if the nerve is handled with care during surgery.

In the author’s presented case, some key surgical points should be addressed:A close CBCT evaluation before surgery;Elimination of all possible inflammation sources in the oral cavity and pre-surgical hygienization;Good local anesthetic injection into the operation can improve tissue preparation;Local anesthetic injection in the nerve should be avoided;Appropriate visibility to the operated field with retractors and periosteal elevators;Surgical field irrigation with saline for improved visibility;Identification of the nerve in the subperiosteal plane grants more nerve safety;Traction of the adjacent tissues is safer than only nerve traction;MN compression should be limited when retractors are positioned on the bone and not on the MN fibers;When using high-speed drills, good irrigation and visibility should be used;When suturing the wound, any nerve fibers should be identified to avoid their possible damage;Early intraoral vitamin B or other agents should be used;Close patient monitoring after the surgery to avoid any hematoma or purulent formation in the operating area.

Limitations of the following study include the small number of cyst lesion cases located in between premolar tooths and mental nerve foramen. On the other hand, the presented case’s strong point could indicate that any similar case of a cyst arising from the jaw bones that is closely related to the inferior alveolar or lingula nerve close proximity can be treated with the same surgical, pharmacological, and subperiosteal elevation technique with good outcomes. Furthermore, a take-away message indicates that CBCT greatly helps each surgeon to avoid unnecessary complications, while a gentle subperiosteal tissue elevation followed by an early postsurgical or a 1–2 day presurgical pharmacological approach also might increase the number of unnecessary complications.

## 4. Conclusions

Possible nerve damage in different surgery types is related to the type of approaches used, the nerve location within a cyst or a tumor, and the method of nerve preparation and retraction with surgical hooks or subperiosteal elevators. In each situation, it is not only wise to identify the nerve fibers during the radiological examination before the surgery but also to know the surgical field anatomy and be able to identify each nerve fiber and its location during each procedure. If possible damage or injury to the nerve will happen, it is worth informing the patient about possible decreased nerve function after the surgery. Careful nerve retraction and good postoperative care grant more satisfactory outcomes.

## Figures and Tables

**Figure 1 medicina-59-00711-f001:**
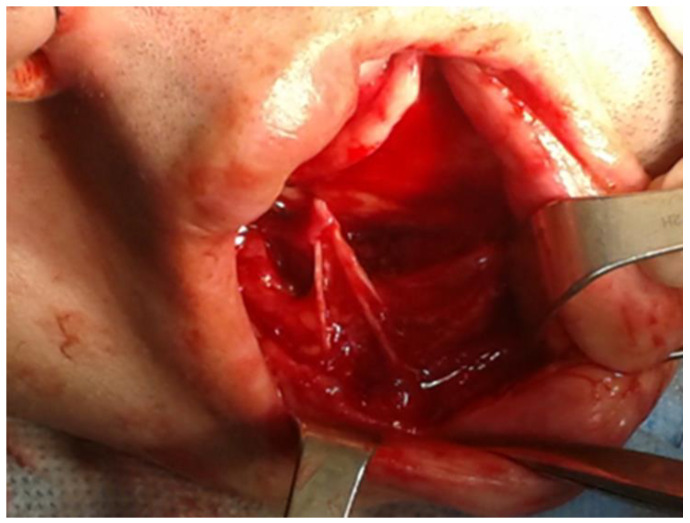
An illustrative figure where the branches of mental nerve are visible during a chin wing osteotomy of the lower border of the mandible during orthognathic surgery.

**Figure 2 medicina-59-00711-f002:**
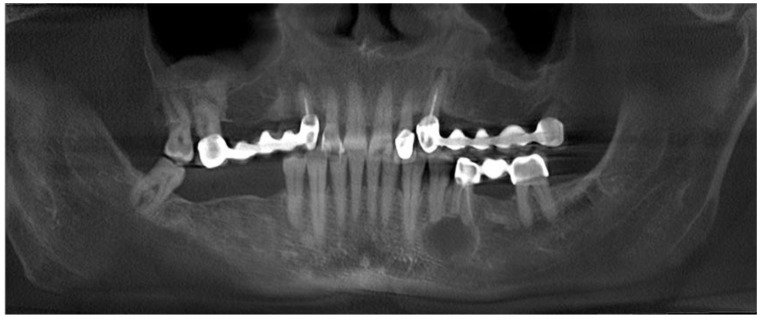
CBCT (cone-beam computed tomography)with clearly visible cyst in the 34/35 area and close proximity of the inferior alveolar nerve.

**Figure 3 medicina-59-00711-f003:**
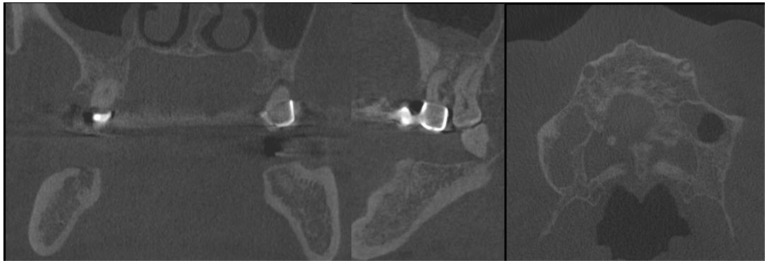
CBCT with visible sources of dental infection. CBCT scans: coronal, sagittal, and axial.

**Figure 4 medicina-59-00711-f004:**
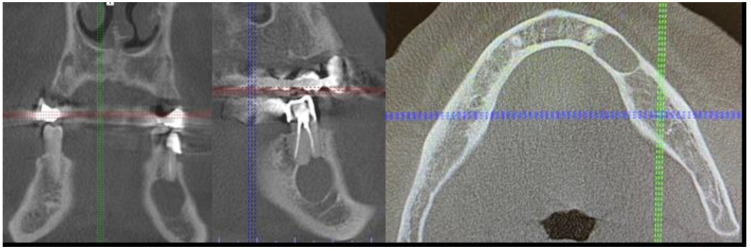
The sagittal, coronal, and axial view of the cystic lesion in the left mandibular basis.

**Figure 5 medicina-59-00711-f005:**
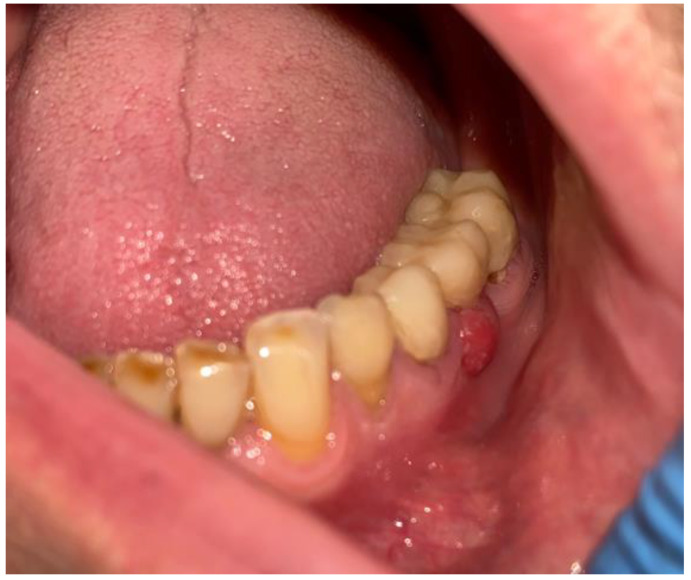
Intraoral photograph with granulation tissues arising from under the dental bridge.

**Figure 6 medicina-59-00711-f006:**
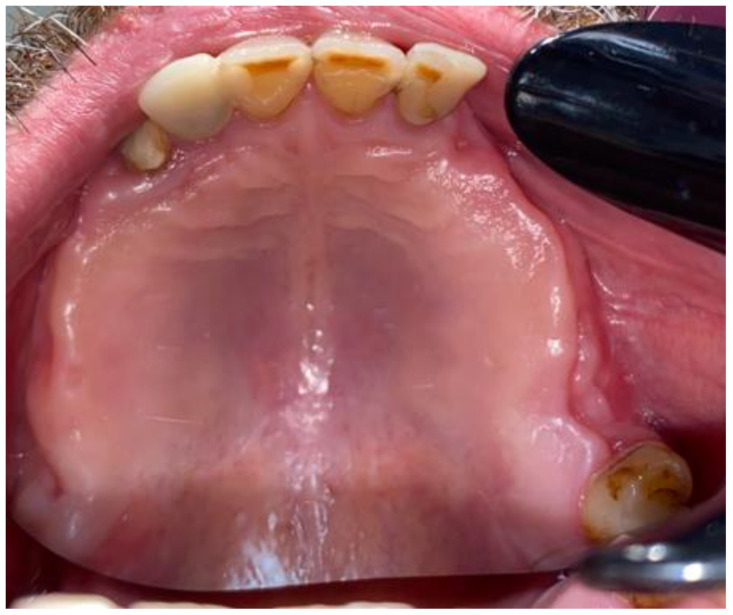
First stages of treatment.

**Figure 7 medicina-59-00711-f007:**
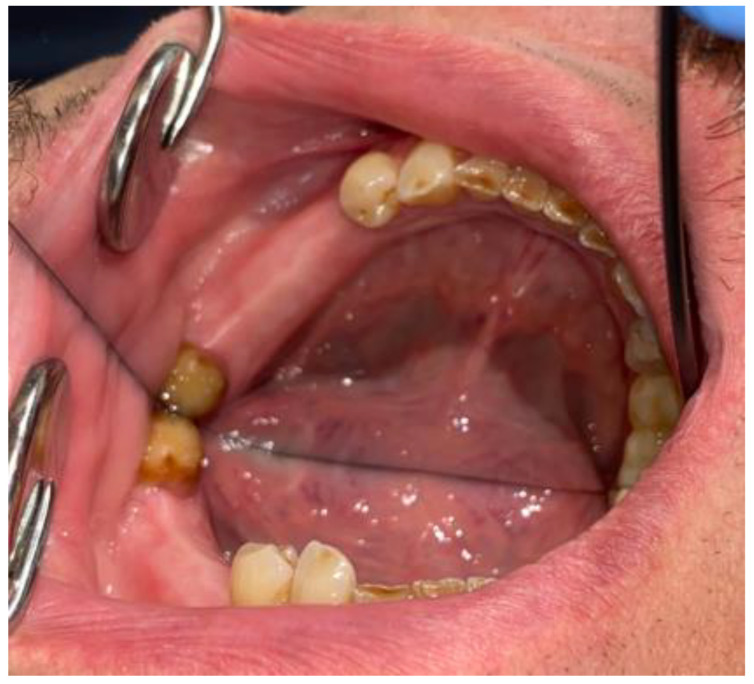
Visible scope of missing teeth scheduled for dental implant placement.

**Figure 8 medicina-59-00711-f008:**
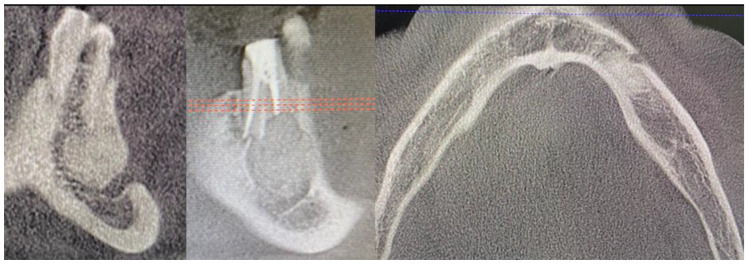
CBCT just after the cyst removal. CBCT presented with the following scans, from the left: coronal, sagittal, and axial. A close proximity to the left mental nerve is seen. Some image disturbances from 3D evaluation are visible.

**Figure 9 medicina-59-00711-f009:**
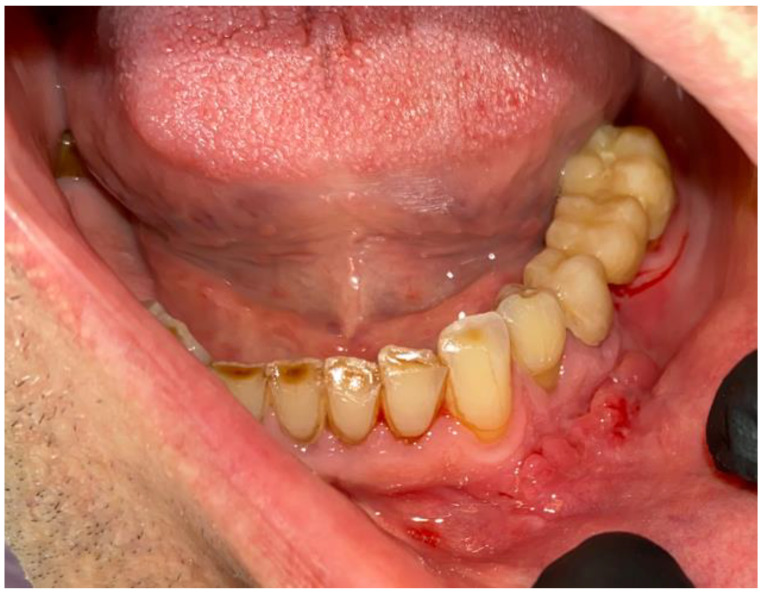
Final view of the operated area after 14 days during suture removal.

**Figure 10 medicina-59-00711-f010:**
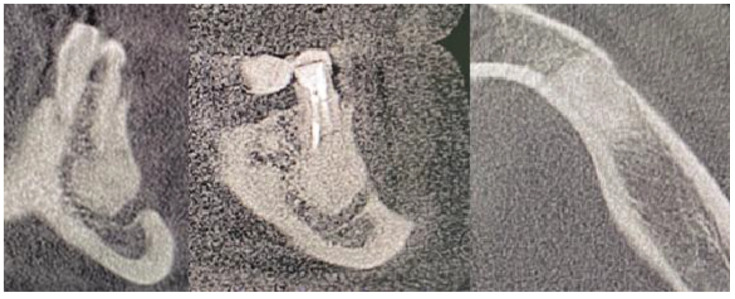
Control CBCT evaluation 1.5 years after a cyst removal. Presented from the left on coronal, sagittal and axial scans. Some image disturbances from 3D evaluation are visible.

**Figure 11 medicina-59-00711-f011:**
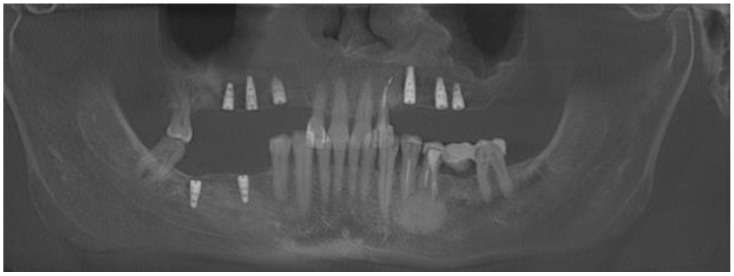
Final result after two years of treatment and dental implant placement.

## Data Availability

Availability of supporting data: the datasets used and/or analyzed during the current study are available from the corresponding author on reasonable request.

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
