# Peer review of "The Temporary Mental Nerve Paresthesia as an Outcome of Dentigerous Cyst Removal during Preparation for Dental Implant Placement: A Case Report"

_medicina, 2023, doi:10.3390/medicina59040711_

Round 1

Reviewer 1 Report

The case report is a presenting a usual procedure to spare the mental nerve in surgery.

The case report is completed with shear hard work. the authors team did a great job. The paper is of Medicina caliber, that need to be decided by the editor. In my openion it is a pretty straightforward finding and management to a maxillofacial surgeon.

The figures 8 and 10 are not clear..needs improvement.

Author Response

Dear reviewer thank You very much.

Response to Reviewer 1 Comments

Point 1. “ The case report is a presenting a usual procedure to spare the mental nerve in surgery.

Response 1: Please provide your response for Point 1. (in red) thank you, the following case report present Authors own case report with follow-up

Point 2. The case report is completed with shear hard work. the authors team did a great job. The paper is of Medicina caliber, that need to be decided by the editor. In my openion it is a pretty straightforward finding and management to a maxillofacial surgeon.

Response 2: Please provide your response for Point 2. (in red) thank you for you kind words dear reviewer. Im more than happy that the paper is suitable for Medicina. Ofcourse I will do what can be done to improve its content.

Point 3. The figures 8 and 10 are not clear..needs improvement.

Response 3: Please provide your response for Point 3. (in red) thank you. Some slight improvement in shape, size, distance with additional description was added. Those three scans are combined all together in a 3d reconstructin patter so that some background noises are visible.

Reviewer 2 Report

This paper describes a case of mental nerve paresthesia after dentigerous cyst removal. The topic is important but it lacks novelty. The surgical procedure is described carefully and in sufficient detail. However, some comments should be addressed:

1)     A manuscript should be accompanied by a checklist according to CARE guidelines

2)     An abstract should be rewritten with a focus on that particular case instead of general phrases

3)     Language proof-reading is necessary (mainly for introduction and discussion)

4)     There is insufficient information about general health of the patient and prior dental history as well as periodontal status

5)     For how long the antibiotic, NSAIDs and vitamin supplementation were prescribed? 

6)     Insufficient attention is paid to the description of post-operative nerve healing: when first signs of sensitivity appeared? When it was fully restored? Any details will be interesting and important.

7)     Please, highlight what is the novelty of this case? Does it actually bring something new to a scientific literature?

Author Response

Response to Reviewer 2 Comments

Point 1. This paper describes a case of mental nerve paresthesia after dentigerous cyst removal. The topic is important but it lacks novelty. The surgical procedure is described carefully and in sufficient detail. However, some comments should be addressed:

Response 1: Please provide your response for Point 1. (in red) thank you, on behalf of myself and all co-authors I will try to implement all the necessary hints. Thank you

Point 2. 1)     A manuscript should be accompanied by a checklist according to CARE guidelines

Response 1: Please provide your response for Point 2. (in red) thank you, according to CARE; title- a case report ; - key words added; - 3a-case unique added; 3b-symptoms added; 3c-3d added; - 4-uniqueness added ; = 5a- added; same as – a-e- clinical and familiar history of diseases + point 6; next points 7-8 added and described = points9-description of therapeutic intervention was improved = results + followup added / improved – in discussion sections strong, weak points and limitations nad take away message were improved

Point 3. 2)     An abstract should be rewritten with a focus on that particular case instead of general phrases

Response 3: Please provide your response for Point 3. (in red) thank you – abstract was re-arranged.

Point 4. 3)     Language proof-reading is necessary (mainly for introduction and discussion)

Response 4: Please provide your response for Point 4. (in red) thank you – paper send to another person, this time US native. Thank you

Point 5. 4)     There is insufficient information about general health of the patient and prior dental history as well as periodontal status

Response 5: Please provide your response for Point 5. (in red) thank you – added accoridng to CARE guidelines.

Point 6.      For how long the antibiotic, NSAIDs and vitamin supplementation were prescribed? 

Response 6: Please provide your response for Point 6. (in red) thank you – missing data added.

Point7 . 6)     Insufficient attention is paid to the description of post-operative nerve healing: when first signs of sensitivity appeared? When it was fully restored? Any details will be interesting and important.

Response 7: Please provide your response for Point 7. (in red) thank you – added

Point 8. 7)     Please, highlight what is the novelty of this case? Does it actually bring something new to a scientific literature?

Response 8: Please provide your response for Point 8. (in red) thank you – added

Reviewer 3 Report

Comments on Nelke et al:

The aim of this manuscript is to describe the outcomes of a cyst removal from the manidibular basis, also discussing treatment modalities.

This manuscript shows rich content, providing a deep insight for some works: the study is within the journal’s scope, and I found it to be well-written, providing sufficient information. Even if the manuscript provides an organic overview, with a densely organized structure and based on well-synthetized evidence, there are some suggestions necessary to make the article complete and fully readable. For these reasons, the manuscript requires major changes.

Please find below an enumerated list of comments on my review of the manuscript:

INTRODUCTION:

LINE 40: Mandibular bone is described, from the genital apophyses to the mylohyoid line, to the mandibular canal, which houses a branch of the “third pair of nerves”, in three sections, from the chin bone to the mandibular foramen (see, for reference: Bernardi S, Angelone AM, Macchiarelli G. Anatomy in dentistry: From the beginnings to contemporary reality. Clin Anat. 2022;35(6):711-722. doi:10.1002/ca.23869). The authors should highlight this issue, in this introductive section.

LINE 88: CBCT (cone-beam tomography) is nowadays a radiological imaging technique, considered a golden standard for the evaluation of any changes in mandibular and maxillary bones, with significant clinical implications (see, for reference: Varvara G, Feragalli B, Turkyilmaz I, et al. Prevalence and Characteristics of Accessory Mandibular Canals: A Cone-Beam Computed Tomography Study in a European Adult Population. Diagnostics (Basel). 2022;12(8):1885. Published 2022 Aug 4. doi:10.3390/diagnostics12081885).

The main topic is interesting, and certainly of great clinical impact. As regards the originality and strengths of this manuscript, this is a significant contribute to the ongoing research on this topic, as it extends the research field on the description of a case, in which a patient with transient mental nerve injury after the removal of a big radicular cyst, in the left mandibular basis, as preparation for dental implant procedures. Overall, the contents are rich, and the authors also give their deep insight for some works.

As regards the section of methods, there is a specific and detailed explanation for the methods used in this study: this is particularly significant, since the manuscript relies on a multitude of methodological and statistical analysis, to derive its conclusions. The methodology applied is overall correct, the results are reliable and adequately discussed.

The conclusion of this manuscript is perfectly in line with the main purpose of the paper: the authors have designed and conducted the study properly. As regards the conclusions, they are well written and present an adequate balance between the description of previous findings and the results presented by the authors.

Finally, this manuscript also shows a basic structure, properly divided and looks like very informative on this topic. Furthermore, figures and tables are complete, organized in an organic manner and easy to read.

In conclusion, this manuscript is densely presented and well organized, based on well-synthetized evidence. The authors were lucid in their style of writing, making it easy to read and understand the message, portrayed in the manuscript. Besides, the methodology design was appropriately implemented within the study. However, many of the topics are very concisely covered. This manuscript provided a comprehensive analysis of current knowledge in this field. Moreover, this research has futuristic importance and could be potential for future research. However, major concerns of this manuscript are with the introductive section: for these reasons, I have major comments for this section, for improvement before acceptance for publication. The article is accurate and provides relevant information on the topic and I have some major points to make, that may help to improve the quality of the current manuscript and maximize its scientific impact. I would accept this manuscript if the comments are addressed properly.

Author Response

Response to Reviewer 3 Comments

Point 1. The aim of this manuscript is to describe the outcomes of a cyst removal from the manidibular basis, also discussing treatment modalities.

Response 1: Please provide your response for Point 1. (in red) thank you -

Point 2. This manuscript shows rich content, providing a deep insight for some works: the study is within the journal’s scope, and I found it to be well-written, providing sufficient information. Even if the manuscript provides an organic overview, with a densely organized structure and based on well-synthetized evidence, there are some suggestions necessary to make the article complete and fully readable. For these reasons, the manuscript requires major changes.

Response 2: Please provide your response for Point 2. (in red) thank you – changes are added.

Point 3. - INTRODUCTION:LINE 40: Mandibular bone is described, from the genital apophyses to the mylohyoid line, to the mandibular canal, which houses a branch of the “third pair of nerves”, in three sections, from the chin bone to the mandibular foramen (see, for reference: Bernardi S, Angelone AM, Macchiarelli G. Anatomy in dentistry: From the beginnings to contemporary reality. Clin Anat. 2022;35(6):711-722. doi:10.1002/ca.23869). The authors should highlight this issue, in this introductive section.

Response 3: Please provide your response for Point 3. (in red) thank you – added.

Point 4. - LINE 88: CBCT (cone-beam tomography) is nowadays a radiological imaging technique, considered a golden standard for the evaluation of any changes in mandibular and maxillary bones, with significant clinical implications (see, for reference: Varvara G, Feragalli B, Turkyilmaz I, et al. Prevalence and Characteristics of Accessory Mandibular Canals: A Cone-Beam Computed Tomography Study in a European Adult Population. Diagnostics (Basel). 2022;12(8):1885. Published 2022 Aug 4. doi:10.3390/diagnostics12081885).

Response 4: Please provide your response for Point 4. (in red) thank you – added.

Point 5. The main topic is interesting, and certainly of great clinical impact. As regards the originality and strengths of this manuscript, this is a significant contribute to the ongoing research on this topic, as it extends the research field on the description of a case, in which a patient with transient mental nerve injury after the removal of a big radicular cyst, in the left mandibular basis, as preparation for dental implant procedures. Overall, the contents are rich, and the authors also give their deep insight for some works.

Response 5: Please provide your response for Point 4. thank you

Point 6. As regards the section of methods, there is a specific and detailed explanation for the methods used in this study: this is particularly significant, since the manuscript relies on a multitude of methodological and statistical analysis, to derive its conclusions. The methodology applied is overall correct, the results are reliable and adequately discussed.

Response 6: Please provide your response for Point 6. thank you

Point 7 . The conclusion of this manuscript is perfectly in line with the main purpose of the paper: the authors have designed and conducted the study properly. As regards the conclusions, they are well written and present an adequate balance between the description of previous findings and the results presented by the authors.

Response 7: Please provide your response for Point 7. thank you

Point8 . Finally, this manuscript also shows a basic structure, properly divided and looks like very informative on this topic. Furthermore, figures and tables are complete, organized in an organic manner and easy to read.

Response 8: Please provide your response for Point 8. thank you

Point 9 In conclusion, this manuscript is densely presented and well organized, based on well-synthetized evidence. The authors were lucid in their style of writing, making it easy to read and understand the message, portrayed in the manuscript. Besides, the methodology design was appropriately implemented within the study. However, many of the topics are very concisely covered. This manuscript provided a comprehensive analysis of current knowledge in this field. Moreover, this research has futuristic importance and could be potential for future research. However, major concerns of this manuscript are with the introductive section: for these reasons, I have major comments for this section, for improvement before acceptance for publication. The article is accurate and provides relevant information on the topic and I have some major points to make, that may help to improve the quality of the current manuscript and maximize its scientific impact. I would accept this manuscript if the comments are addressed properly

Response 9: Please provide your response for Point 9. thank you

Round 2

Reviewer 2 Report

Dear authors,

I am satisfied with the corrections made.

Reviewer 3 Report

Manuscript can be now accepted